# Protective Effect of NO_2_-OA on Oxidative Stress, Gliosis, and Pro-Angiogenic Response in Müller Glial Cells

**DOI:** 10.3390/cells12030494

**Published:** 2023-02-02

**Authors:** María V. Vaglienti, Paula V. Subirada, Mariana B. Joray, Gustavo Bonacci, María C. Sánchez

**Affiliations:** 1Departamento de Bioquímica Clínica, Facultad de Ciencias Químicas, Universidad Nacional de Córdoba, Córdoba 5000, Argentina; 2Centro de Investigaciones en Bioquímica Clínica e Inmunología (CIBICI), Consejo Nacional de Investigaciones Científicas y Técnicas (CONICET), Córdoba 5000, Argentina; 3Instituto de Investigación Médica Mercedes y Martín Ferreyra (INIMEC), Consejo Nacional de Investigaciones Científicas y Técnicas (CONICET), Córdoba 5000, Argentina; 4Facultad de Ciencias Químicas, Universidad Católica de Córdoba, Córdoba 5000, Argentina; 5Instituto de Investigaciones en Recursos Naturales y Sustentabilidad José Sánchez Labrador J. S., Consejo Nacional de Investigaciones Científicas y Técnicas (CONICET), Córdoba 5000, Argentina

**Keywords:** Müller glial cells, nitro-oleic acids, Nrf2-Keap1 antioxidant pathway, gliosis, pro-angiogenic factor, reactive oxygen species

## Abstract

Inflammation and oxidative and nitrosative stress are involved in the pathogenesis of proliferative retinopathies (PR). In PR, a loss of balance between pro-angiogenic and anti-angiogenic factors favors the secretion of vascular endothelial growth factor (VEGF). This vascular change results in alterations in the blood–retinal barrier, with extravasation of plasma proteins such as α_2_-macroglobulin (α_2_M) and gliosis in Müller glial cells (MGCs, such as MIO-M1). It is well known that MGCs play important roles in healthy and sick retinas, including in PR. Nitro-fatty acids are electrophilic lipid mediators with anti-inflammatory and cytoprotective properties. Our aim was to investigate whether nitro-oleic acid (NO_2_-OA) is beneficial against oxidative stress, gliosis, and the pro-angiogenic response in MGCs. Pure synthetic NO_2_-OA increased HO-1 expression in a time- and concentration-dependent manner, which was abrogated by the Nrf2 inhibitor trigonelline. In response to phorbol 12-myristate 13-acetate (PMA) and lipopolysaccharide (LPS), NO_2_-OA prevented the ROS increase and reduced the gliosis induced by α_2_M. Finally, when hypoxic MGCs were incubated with NO_2_-OA, the increase in VEGF mRNA expression was not affected, but under hypoxia and inflammation (IL-1β), NO_2_-OA significantly reduced VEGF mRNA levels. Furthermore, NO_2_-OA inhibited endothelial cell (BAEC) tubulogenesis. Our results highlight NO_2_-OA’s protective effect on oxidative damage, gliosis; and the exacerbated pro-angiogenic response in MGCs.

## 1. Introduction

Proliferative retinopathies (PR), such as retinopathy of prematurity (ROP) and proliferative diabetic retinopathy (PDR), are the major cause of vision loss in the infant and adult populations, respectively [1]. Although the etiology and pathogenesis are different for these retinopathies, they clinically present two main stages. The first stage is associated with vascular obliteration and growth cessation, leading to retinal hypoxia, while the second stage, or neovascularization, is characterized by unfunctional deregulated vascular growth to compensate for the hypoxic state of the retina [2]. During this stage, a loss of balance occurs between pro- and anti-angiogenic factors, mainly due to an increase in vascular endothelial growth factor (VEGF) in the retinal tissue [3]. Although numerous types of cells in the retina secrete VEGF, it has been reported that Müller glial cells (MGCs) are the principal source of this pro-angiogenic factor in PR [4].

It is well-known that MGCs play an important role in healthy and sick retinas, including PR [5]. MGCs become reactive in response to harmful stimuli to protect retinal function. This activation process, or gliosis, in MGCs induces the expression of intermediate filaments such as glial fibrillary acidic protein (GFAP) and vimentin. In PR, the activation of gliosis has been described as protective at the beginning of pathology, but if it persists over time the consequences can be detrimental for retinal function [6].

α_2_-Macroglobulin (α_2_M) is a multifunctional protease inhibitor present in plasma in its native form or as a complex between protein inhibitor and protease, called the active form (α_2_M*). In both forms, α_2_M has been described as participating in the transport of different growth factors and cytokines as well as other physiological effectors [7]. Due to those properties, α_2_M can influence a broad range of key biological processes through its interaction with the receptor LRP1 or by binding to cytokines and growth factors [8]. In the retina, we previously demonstrated that α_2_M* induces gliosis in MGCs at similar concentrations to those present in the vitreous humor of patients with retinal diseases [9].

In addition, MGCs in PR participate in the secretion of inflammatory cytokines/chemokines, contributing to retinal neovascularization [10,11]. This action favored the induction of oxidative stress and the pro-inflammatory state, which play a key role in the subsequent retina damage, including vascular alterations, gliosis, and neurodegeneration [12]. Thus, the development of new therapies for PR should focus on preventing either vascular or neuronal alterations.

Nitrated fatty acids (NO_2_-FAs) are important endogenous electrophilic signaling mediators generated through the reaction of unsaturated fatty acids with nitric oxide (NO) and NO-derived species [13,14,15]. The NO_2_-FA or nitroalkene family includes nitro-oleic acids (NO_2_-OAs), nitro-linoleic acids (NO_2_-LAs), and nitro-conjugated linoleic acids (NO_2_-CLAs). NO_2_-FAs are present in the plasma and urine of healthy individuals at low micromolar concentrations [16,17] and can be modulated during inflammation, as was demonstrated in a mouse model of cardiac ischemia and reperfusion [18,19]. NO_2_-FAs induce post-translational modification through Michael addition reactions with susceptible nucleophiles, such as cysteine, in key metabolic proteins [20,21]. Thus, they drive the activation or inhibition of signaling pathways related to antioxidant or anti-inflammatory actions such as Keap1/Nrf2 and NF-κB, among others [21,22]. These reactions are critical for the anti-inflammatory and antioxidant responses induced by NO_2_-FA in different mouse models of atherosclerosis, ischemia and reperfusion injury, diabetes, and metabolic syndrome.

Since it is widely described that the induction of the retinal antioxidant defense system through the activation of the Keap1/Nrf2 signaling pathway improves the resolution and prevention of PR [23,24,25,26], we hypothesized that NO_2_-OA could be beneficial against oxidative stress, gliosis, and the pro-angiogenic response in MGCs. Moreover, we evaluated the effect of NO_2_-OA on tubulogenesis in endothelial cells (ECs).

In the present work, we show that NO_2_-OA induced HO-1 expression in a time- and concentration-dependent manner and prevented damage induced by PMA, LPS, and α_2_M* on MGCs. Furthermore, we observed that NO_2_-OA avoided the gliosis induced by α_2_M*. In addition, NO_2_-OA did not modulate the expression of VEGF-A under hypoxia. However, under pro-inflammatory conditions, NO_2_-OA significantly reduced VEGF-A expression in MGCs. Finally, we demonstrated that NO_2_-OA inhibited EC tube formation, while treatment with trigonelline abrogated that effect. Thus, these findings underscore the protective role of NO_2_-OA against oxidative stress, gliosis, and the pro-angiogenic response in MGCs, preventing the development of newly formed blood vessels.

In conclusion, these data suggest a potential therapeutic application of NO_2_-OA for the treatment of PR in order to attenuate both the vascular and non-vascular components.

## 2. Materials and Methods

### 2.1. Materials

A spontaneously immortalized human Müller glial cell line (MIO-M1) was kindly provided by G. Astrid Limb (UCL Institute of Ophthalmology and Moorfields Eye Hospital, London, UK). A bovine aortic endothelial cell line (BAEC) was purchased from the American Type Culture Collection (ATCC^®^) (Manassas, VA, USA). Nitro-oleic acid (NO_2_-OA) and Oleic acid (OA) were provided by Bruce A. Freeman and Francisco Schopfer (Department of Pharmacology and Chemical Biology, University of Pittsburgh, Pittsburgh, PA, USA). Lipopolysaccharide (LPS), phorbol 12-myristate 13-acetate (PMA), and a 2′,7′-dichlorofluorescein diacetate probe (2′,7′-DCFH-DA) were purchased from Sigma. α_2_M was purified from human plasma using a previously described procedure [27]. α_2_M was activated (α_2_M*) by incubation with 200 mM methylamine-HCl for 6 h at pH 8.2, as previously described [28]. Recombinant mouse IL-1 beta/IL-1F2 protein was obtained from R&D Systems Biotechne. Suramin sodium was purchased from Santa Cruz Biotechnology (Dallas, TX, USA). Extracellular matrix (ECM, Matrigel^®^) was purchased from Corning Incorporated (Glendale, AZ, USA). Recombinant human vascular endothelial growth factor (VEGF) 165 was purchased from BPSBiosciences (6405 Mira Mesa Blvd. Suite 100, San Diego, CA 92121, USA).

### 2.2. Cell Line and Culture Reagents

MIO-M1 cells were grown in complete high-glucose Dulbecco’s modified Eagle’s medium (DMEM; Invitrogen, Buenos Aires, Argentina) containing 10% fetal bovine serum (FBS), 2 mM L-glutamine (GlutaMAX; Invitrogen), and 50 U/mL penicillin/streptomycin (Invitrogen). BAEC were grown in complete high-glucose DMEM containing 20% FBS, 2 mM L-glutamine, 50 U/mL penicillin/streptomycin, and 0,03 mg/mL of an endothelial cell growth supplement from a bovine pituitary gland (Sigma). A sterile plastic material was obtained from Greiner Bio-One (Frickenhausen, Germany). Cells were maintained in a 5% CO_2_ humidified culture incubator at 37 °C.

### 2.3. Cell Viability Assay

MIO-M1 cell viability was evaluated using the metabolic dye 3-(4,5-dimethlthiazol-2-yl)-2,5-diphenyl-tetrazolium bromide (MTT). Briefly, 2 × 10^3^ cells/well were seeded in a 96-well plate and treated with vehicle (methanol) or different doses of NO_2_-OA (0.1 to 10 µM) for 24 h and 72 h. Then, 10 μL of the yellow tetrazolium salt MTT (5 mg/mL) was added to the culture medium. Cells were incubated with the MTT reagent for an additional 3 h at 37 °C. After that, the cell medium was carefully removed, and 200 μL of DMSO was added in order to solubilize the crystal violet. Finally, the optical density values were measured at a wavelength of 570 nm using a SpectraMax M5 plate reader (Molecular Devices,CA, USA). The results are expressed as a percentage of cell viability relative to control.

### 2.4. Western Blot Assay

MIO-M1 cells (2 × 10^5^/well) were seeded on a six-well plate in complete media and serum-deprived media for 2 h prior to all treatments. All incubations were performed in DMEM containing 2% FBS. NO_2_-OAs are solubilized in methanol and added to the cultures at a 1/1000 dilution in DMEM for all treatments. MIO-M1 cells were treated with vehicle (methanol) or different concentrations of NO_2_-OA (0.1, 1.0, 2.5, or 5.0 µM) for 8 or 16 h. On the other hand, MIO-M1 cells were incubated in the presence or absence of 5 µM of NO_2_-OA for 30 min before adding α_2_M* (60 nM) for 2, 4, or 6 h. Cells were lysed in 1% Triton X-100 in PBS supplemented with 1 mM phenylmethylsulphonyl fluoride (PMSF), 10 mM sodium ortho-vanadate, and protease and phosphatase inhibitor cocktails (Sigma-Aldrich). BCA protein assay kits were used for protein quantification (Pierce BCA, Thermo Scientific, MA, USA). Total cell lysates (20 µg) were resolved using 10% SDS-PAGE and transferred onto nitrocellulose membranes (Amersham Hybond ECL; GE Healthcare Bio-Sciences AB, Uppsala, Sweden). Membranes were blocked with 5% milk in TBS containing 0.1% Tween-20 (TBST) for 1 h at room temperature (RT). Membranes were probed with specific primary antibodies according to the antibody manufacturer’s specifications and normalized using housekeeping proteins (β-actin). The following primary antibodies were used: rabbit anti-HO-1 (1/1000; Enzo Life Science), rabbit polyclonal anti-GFAP (1/1000; Dako, Carpinteria, CA, USA), mouse monoclonal anti-vimentin (1/1000; M7020, Dako), and mouse monoclonal anti-β-actin (1/2000; ab8226, Abcam). The secondary antibodies were IRDye 800 CW donkey anti-rabbit IgG and IRDye 800 CW donkey anti-mouse IgG antibodies (1:15,000 in 1% BSA TBST). Membranes were visualized and quantified using the Odyssey Infrared Imaging System (LI-COR, Inc., Lincoln, NE, USA).

### 2.5. Dichlorofluorescein Assay

The ROS production in MIO-M1 cells was quantified using the dichlorofluorescein (DCF) assay, following a previously described procedure [29]. Briefly, MIO-M1 cells were treated with NO_2_-OA (5 μM) or vehicle (methanol) for 6 h before adding PMA (1 µg/mL) or LPS (1 nM) for 30 min. The stocks of PMA (1 mg/mL) and LPS (1 μM) were prepared in DMSO and DMEM, respectively. A 2′,7′-dichlorofluorescein diacetate probe (5 μM) was added, and the cells were incubated for 30 min. The DCF fluorescence intensity was measured with an FACSCanto II flow cytometer (BD Biosciences) and analyzed with FlowJo software (ThreeStar, FlowJo eI_V10). Moreover, MIO-M1 cells were treated with or without NO_2_-OA for 30 min before the α_2_M* stimulus for 2, 4, or 6 h, and the ROS levels were determined as above.

### 2.6. Quantitative Real-Time Reverse-Transcription PCR (qRT-PCR)

Total RNA was extracted from cultured cells using Trizol Reagent (Invitrogen) [30]. Briefly, 1 μg of total RNA was reverse-transcribed in a total volume of 20 μL using random primers (Invitrogen) and 50 U of M-MLV reverse transcriptase (Promega Corp.). Then, cDNA was mixed with 1× SYBR Green PCR Master Mix (Applied Biosystems) and forward and reverse primers: VEGF-A forward—CCGCAGACGTGTAAATGTTCCT and VEGF-A reverse—CGGCTTGTCACATCTGCAAGTA. qPCRs were carried out on an Applied Biosystems 7500 Real-Time PCR System with Sequence Detection Software v1.4. The cycling conditions included a hot start at 95 °C for 10 min, followed by 40 cycles at 95 °C for 15 s and 60 °C for 1 min. The specificity was verified using a melting curve analysis. The results were normalized to GAPDH: forward—GATGCCCCCATGTTTGTGAT and reverse—GGTCATGAGTCCTTCCACGAT. The relative gene expression was calculated according to the ^2-ΔΔCt^ method. Each sample was analyzed in triplicate. No amplification was observed in PCRs using water as a template.

### 2.7. Hypoxic Assays

For gas hypoxia, cells were grown at 60–70% confluence in normal conditions and transferred to a gas culture chamber (StemCell Technologies, Vancouver, BC, Canada) supplied with 1% O_2_, 94% N_2_, and 5% CO_2_. Control cells were kept in normoxia (21% O_2_). Cell experiments were conducted for 24 h, as previously described [31].

### 2.8. Tube Formation Assay

This assay was performed according to Arnaoutova and Kleinman [32]. Briefly, BAECs (≈1.5 × 10^4^ cells) were placed on a 96-well half area plate previously coated with 30 µL of Matrigel. The plates were incubated for 18 h (37 °C and 5% CO_2_) in the presence or absence of NO_2_-OA (5 µM) and trigonelline (1 µM), with the addition of VEGF (10 ng/mL) as an angiogenic stimuli. NO_2_-OA and trigonelline stock were prepared in DMSO and PBS, respectively. Recombinant VEGF was reconstituted in distilled water to a concentration of 0.1 mg/mL. Suramin sodium (30 µM) was used as a positive control of inhibition. The stock solution of Suramin sodium was prepared in ethanol. Controls with or without vehicles (DMSO or ethanol) were run simultaneously. The final concentration of vehicle in all experiments described herein was 0.1% *v*/*v*. No adverse effects were observed at this concentration. The images were obtained with an Olympus CKX41 inverted microscope and analyzed with the software ImageJ. The tubular structures were quantified, and the percentages of inhibition (I%) were calculated as follows: I% = [1 − (total tube length treatment/total tube length control)] × 100.

### 2.9. Statistical Analysis

The statistical analysis was performed using the GraphPad Prism 7.0 software. After confirmation of variance homogeneity evaluated by F or Bartlett’s tests, the data were analyzed using a one-way analysis of variance (ANOVA), followed by Dunnett’s multiple comparison post-test to determine statistical significance among more than two different groups. A *p*-value < 0.05 was considered statistically significant. Means ± standard errors (SEMs) are shown in the graphs.

## 3. Results

### 3.1. NO_2_-OA Induces HO-1 Expression in MGCs

Previous studies of NO_2_-FAs have shown the activation of the Keap1/Nrf2 antioxidant pathway in ECs [33] and macrophages [34] as well as in renal and cardiac tissues, among others. Since the activation of Keap1/Nrf2 has cytoprotective and anti-inflammatory effects in retinal tissue [23,24,25,26], we were interested in evaluating NO_2_-OA’s effect on MGCs. Therefore, we analyzed MGC viability under NO_2_-OA treatment using an MTT assay. Exposing MIO-M1 cells to vehicle (methanol) or NO_2_-OA (0.1–10 μM) for 24 to 72 h did not show a statistically significant reduction in cell viability (*p* > 0.05) compared to control cells (untreated) (Figure 1A). As a control non-electrophilic fatty acid, OA was used (10 μM). Then, we studied whether NO_2_-OA induced a change in the expression of the Keap1/Nrf2 downstream target gene (HO-1) at 8 and 16 h (Figure 1B,C). While NO_2_-OA significantly increased HO-1 expression in a time- and concentration-dependent manner, no changes in the expression were observed with OA in MIO-M1 cells (Figure 1B,C). Regarding HO-1 expression, we observed the highest increases with a 5 µM NO_2_-OA treatment (Figure 1D), where 46.9- and 8.4-fold increases were described at 8 and 16 h, respectively (Figure 1D). In addition to protein expression, NO_2_-OA induced the transcriptional expression of HO-1 mRNA in MIO-M1 cells (Figure 1E). OA was unable to induce changes in HO-1 at the transcriptional level. To evaluate the participation of Keap1/Nrf2 in the activation of HO-1 by NO_2_-OA, we used the Nrf2 inhibitor trigonelline. The induction of HO-1 mRNA expression by NO_2_-OA was significantly reduced in MIO-M1 cells treated with trigonelline compared to control (Figure 1F). Interestingly, trigonelline was unable to block the complete effect of NO_2_-OA over HO-1. Taken together, these results show that NO_2_-OA treatment did not affect MGC viability and induced increases in HO-1 expression at 8 and 16 h via Nrf2 in MIO-M1 cells.

### 3.2. NO_2_-OA Prevents the Increase in ROS Levels Induced by LPS or PMA in MGCs

Previous studies have demonstrated that oxidative stress and inflammation drive the vascular injury and neuronal dysfunction described in PR [12,35]. Since the accumulation of ROS promotes detrimental modifications in proteins, lipids, and DNA, leading to cellular stress or cell death, we examined whether NO_2_-OA could protect retinal cells exposed to oxidative stress. For this purpose, we measured ROS with the probe DCF-DA using flow cytometry in MGCs incubated with 5 µM NO_2_-OA prior to the addition of PMA (1 µM) or LPS (1 µg/mL) for 30 min. PMA and LPS significantly increased ROS levels in MIO-M1 cells compared to control (*p* < 0.001). However, the NO_2_-OA treatment prevented ROS induction by PMA or LPS (*p* > 0.05) (Figure 2A,B). Overall, these results demonstrate the protective role of NO_2_-OA against oxidative stress in MGCs.

### 3.3. NO_2_-OA Reverts Glial Stress Induced by α_2_M* in MGCs

Alterations in the blood–retinal barrier in PR allow the extravasation of plasma proteins [36]. Furthermore, we previously demonstrated that α_2_M* induces gliosis in MGCs, as judged by the expression of GFAP in in vivo and in vitro models [9]. In order to evaluate whether NO_2_-OA abrogates glial stress in MGCs, we incubated MIO-M1 cells in the presence or absence of 5 µM NO_2_-OA for 30 min before α_2_M* (60 nM) treatment for 2, 4, or 6 h. α_2_M* significantly increased GFAP and vimentin and affected HO-1 expression (Figure 3A). A densitometry analysis showed that α_2_M* induced 2.7-, 2.6-, and 3.3-fold changes in the expression of GFAP at 2, 4, and 6 h, respectively, compared to control (Figure 3B). Similar results were described for vimentin, which reached its highest expression at 2 h. NO_2_-OA reduced the expression of the glial stress marker induced by α_2_M* in MIO-M1 cells, but this reduction occurred after 6 h of incubation with α_2_M*. Thus, NO_2_-OA prevented the harmful consequences of persistent gliosis in MGCs. As a control for NO_2_-OA activity, we evaluated the HO-1 expression in MIO-M1 cells (Figure 3B). In summary, these data indicate that α_2_M*-dependent gliosis was reduced by NO_2_-OA as a result of the attenuation of GFAP and vimentin expression in MGCs.

### 3.4. NO_2_-OA Inhibits ROS Induced by α_2_M* in MGCs

Given that α_2_M* induces gliosis, we evaluated whether α_2_M* could be involved in ROS generation and its participation in glial stress in MGCs. Hence, we measured the ROS levels in MIO-M1 cells stimulated with α_2_M* (60 nM) at 2, 4, or 6 h with the DCF-DA probe using flow cytometry. A quantitative analysis of fluorescence (FITC-A) geometric mean values showed that α_2_M* significantly increased (1.7, 2.1, and 2.9-fold) ROS levels in a time-dependent manner (Figure 4A). To gain insight into NO_2_-OA’s protective role against the gliosis induced by α_2_M*, we evaluated whether NO_2_-OA could prevent the ROS induced by α_2_M*. For this purpose, MIO-M1 cells treated with or without 5 µM NO_2_-OA for 30 min were incubated with α_2_M* (60 nM) at 2, 4, or 6 h. (Figure 4B). The results indicate that NO_2_-OA ameliorated the ROS levels induced by α_2_M* at 4 and 6 h (Figure 4B). On the other hand, as a control, we determined that NO_2_-OA induced a slight reduction in the basal ROS levels at the evaluated experimental times (Figure 4C). These findings indicate that the induction of ROS by α_2_M* may be a precursor of gliosis and confirm the antioxidant effect of NO_2_-OA in MGCs.

### 3.5. Effect of NO_2_-OA on Angiogenesis in MGCs and ECs

Since MGCs are the main regulator of angiogenesis in PR through synthesizing and secreting the pro-angiogenic factor VEGF [4,31,37], we evaluated whether NO_2_-OA could modulate the secretion of VEGF in MGCs. In order to mimic the hypoxic and pro-inflammatory conditions of PR, we treated MIO-M1 cells under normoxia (21% O_2_) or hypoxia (1% O_2_) in the presence or absence of 5 µM NO_2_-OA and IL-1β (20 ng/mL) for 24 h. The qRT-PCR results revealed that hypoxia induced a significant increase in the VEGF-A mRNA levels in MIO-M1 cells (12.2-fold). This increase in VEGF-A mRNA expression was exacerbated in the presence of the pro-inflammatory IL-1β (45.5-fold) (Figure 5A). The NO_2_-OA treatment was unable to regulate the VEGF mRNA induced by hypoxia (16.5-fold), but NO_2_-OA was efficient in reducing VEGF mRNA expression in the hypoxic and inflammatory condition from a 45.5- to a 26.0-fold increase (Figure 5A). On the other hand, when MIO-M1 cells were exposed under normoxic or normoxic/inflammatory conditions, NO_2_-OA did not affect VEGF-A mRNA expression (Figure 5A). Thus, these results indicate that NO_2_-OA could modulate VEGF-A expression in an inflammatory milieu but was unsuccessful in reverting VEGF-A expression induced under hypoxic conditions in MGCs.

ECs have a leading role in angiogenesis during PR [4]. Therefore, we evaluated whether the NO_2_-OA treatment could have a direct impact on tubulogenesis. To assess the effect of NO_2_-OA on EC tube formation, we seeded BAECs in Matrigel and treated them with vehicle (methanol) or 5 µM NO_2_-OA in the presence of VEGF (10 ng/mL). NO_2_-OA induced a significant reduction in EC tube formation compared to control cells (Figure 5B). A quantitative analysis showed that NO_2_-OA significantly decreased the mesh number (polygonal vascular structures), average mesh area, nodes, and total segment length in the evaluated conditions (Figure 5C). These changes were not observed with the vehicle, and in addition, a control of the inhibition of tubulogenesis with sodium suramin was used (30 µM). In concordance with these findings, NO_2_-OA induced the inhibition of tube formation with respect to VEGF control in BAECs (Figure 5D).

To assess whether the anti-angiogenic effect of NO_2_-OA was mediated by the activation of the Keap1/Nrf2 antioxidant pathway, we used trigonelline in the experimental design. A quantitative analysis showed that 1 µM trigonelline restored the number of polygons and nodes and the total segment length to control levels (Figure 5C,D). Taken together, these results suggest that the NO_2_-OA modulation of retinal vasoproliferative events may be through the activation of the Keap1/Nrf2 pathway.

## 4. Discussion

Pathological retinal angiogenesis is a leading cause of vision loss in eye diseases such as PDR and ROP [1,38]. Accumulating evidence from experimental models and clinical studies of PR have provided knowledge related to the vascular injury [39]. However, it is known that glial cells and neurons are also altered [40,41]. Among the converging etiopathogenic mechanisms are inflammation, oxidative damage, and the increased production of reactive nitrogen species [42,43]. High levels of ROS cause cytotoxicity or apoptosis, but they have also been established as second messengers regulating retinal VEGF levels [44,45]. In fact, several studies have been carried out in order to reduce the excessive ROS production in retinopathies [46,47]. However, an unexplored therapeutic strategy in the retina is the modulation of transcription factors involved in antioxidant defense, such as the Keap1/Nrf2 system, which regulates the expression of HO-1 and NQO1 as well as enzymes involved in glutathione biosynthesis, among others. Recently, numerous studies have demonstrated the anti-inflammatory and cytoprotective effects of NO_2_-FAs [21,22,48]. Here, we investigated the effects of NO_2_-OA on the oxidative stress, gliosis, and pro-angiogenic response in MGCs.

Initially, we evaluated if the NO_2_-OA treatment was able to induce the expression of detoxifying enzymes such as HO-1 in MGCs. Our results showed that NO_2_-OA induced HO-1 at the protein and transcriptional levels in MIO-M1 cells by activating the Keap1/Nrf2 antioxidant pathway. To corroborate this result, we used the pharmacological inhibitor trigonelline [49]. As previously described by Arlt et al. [49], the inhibition of Nrf2 by the trigonelline treatment was concentration-dependent, showing the greatest inhibition at concentrations between 0.1 and 1 μM. In line with this result, we showed that 1 μM trigonelline had significantly higher inhibitory effects on the activation of Nrf2 than 25 μM. Interestingly, trigonelline did not reduce HO-1 expression to control levels, indicating that NO_2_-OA might act in either Nrf2-dependent or -independent manners on the modulation of HO-1. This result may be attributed to previous work from Wright et al., which showed the transcriptional modulation of HO-1 by NO_2_-LA (nitro linoleic acid) through cAMP, AP-1, and E-box response element interactions in MEF and human aortic endothelial cells [50].

Oxidative stress is known to be associated with PR [45,51,52], and under these circumstances MGCs release antioxidant molecules [40] to maintain the redox balance in the retinal tissue. In the present work, as expected, MIO-M1 cells exposed to LPS or PMA significantly triggered intracellular ROS production, which was decreased by NO_2_-OA. This indicates that the level of antioxidant enzymes was sufficient to effectively prevent the increase in ROS triggered by LPS and PMA. A similar effect has been reported in bone-marrow-derived macrophages isolated from ApoE^-/-^ mice, showing attenuated values of superoxide production [53].

It is widely known that MGCs, in response to a stressor, are able to induce persistent gliosis in PR [6,31,37]. In previous studies, we showed that α_2_M*, in concentrations similar to those found in the vitreous humor of subjects with PR [54], increased GFAP levels in accordance with MGC activation [9]. Our results demonstrate that the α_2_M* treatment significantly increased the protein expression of GFAP and vimentin. In addition, a slight increase in HO-1 expression was also observed, which may be associated with the induction of ROS and gliosis by α2M*. Likewise, the NO_2_-OA treatment prevented the expression of GFAP and vimentin induced by α_2_M*. However, the strong antioxidant response was only observed at longer times.

There is indirect evidence in the literature that ROS play a role in *gliosis *[55,56,57]. To add further complexity to the study, we next evaluated whether α_2_M* could also increase the ROS and if this increase could subsequently induce gliosis in MGCs. Interestingly, α_2_M* seems to increase ROS in a time-dependent manner, and NO_2_-OA treatment modulates ROS to basal levels. These results suggest a relationship between ROS levels and gliosis induced by α_2_M*. Further studies are needed to determine the exact mechanism.

Inflammation-associated angiogenesis also occurs during pathophysiological events such as PR [35,58]. Therefore, we decided to investigate the participation of MGCs under hypoxic and pro-inflammatory conditions in VEGF-mediated vasoproliferative events. IL-1β is a pro-inflammatory cytokine that is widely involved in different diseases [59]. Accordingly, it has been described as a key player in inflammatory episodes leading to preterm labor [60] and consequently to premature babies with a risk of ROP [61]. Previous studies have shown that IL-1β induces both the transcriptional activation [62,63] and increased stability of VEGF mRNA by different mechanisms, including NF-κB [64]. Knowing that NO_2_-OA inhibits this pro-inflammatory pathway, we decided to evaluate the expression of VEGF mRNA in MIO-M1 cells treated with IL-1β as well as the effect of NO_2_-OA on normoxic or hypoxic conditions. As expected, a significant increase in VEGF mRNA levels under hypoxic conditions was observed, but the NO_2_-OA treatment failed to modulate the VEGF levels. However, a combination of IL-1β and hypoxia induced a synergic effect on VEGF levels that was inhibited by NO_2_-OA. Similarly, under normoxic conditions, VEGF showed a slight increase after IL-1β stimulation, which was also inhibited by NO_2_-OA. Overall, these results suggest that NO_2_-OA exerts a protective modulation on the induction of this pro-angiogenic factor with IL-1β in MGCs, probably by inhibiting the NF-κB pathway. Further studies should be conducted to evaluate this anti-angiogenic mechanism.

On the other hand, Wei et al. [65] demonstrated that Nrf2 is a critical regulator of angiogenesis in the development of the mammalian retina through the modulation of VEGF in tip cell formation and vascular branching [66]. Furthermore, Huang et al. [24] recently reported that nattokinase inhibits VEGF-induced tube formation via Nrf2 activation. Accumulating evidence stemming from experimental models and clinical studies have provided insights into the mechanisms of vascular injury leading to pathological vitreoretinal NV and the implementation of ocular anti-VEGF therapies [39]. Bevacizumab (Avastin^®^; Genentech, San Francisco, CA, USA), ranibizumab (Lucentis^®^; Genentech, San Francisco, CA, USA), and aflibercept (Eylea^®^; Regeneron Pharmaceuticals, Tarrytown, NY, USA) are the most widely used anti-VEGF drugs administered by intravitreal injection in ocular pathologies. In that sense, we assessed the effect of NO_2_-OA on angiogenesis using an in vitro tubulogenesis assay. NO_2_-OA strongly inhibited the formation of tubular structures, while an Nrf2 inhibitor reversed this effect, demonstrating that NO_2_-OA regulates angiogenesis via an Nrf2-dependent pathway.

In summary, our results constitute the first evidence of the beneficial and protective effects of NO_2_-OA in modulating the oxidative stress, gliosis, and pro-angiogenic response associated with inflammation in MGCs. Additional experiments are in progress in an oxygen-induced-retinopathy mouse model to highlight the clinical and therapeutic connotations of our in vitro results in neovascular and neurodegenerative disorders of the retina.

## Figures and Tables

**Figure 1 cells-12-00494-f001:**
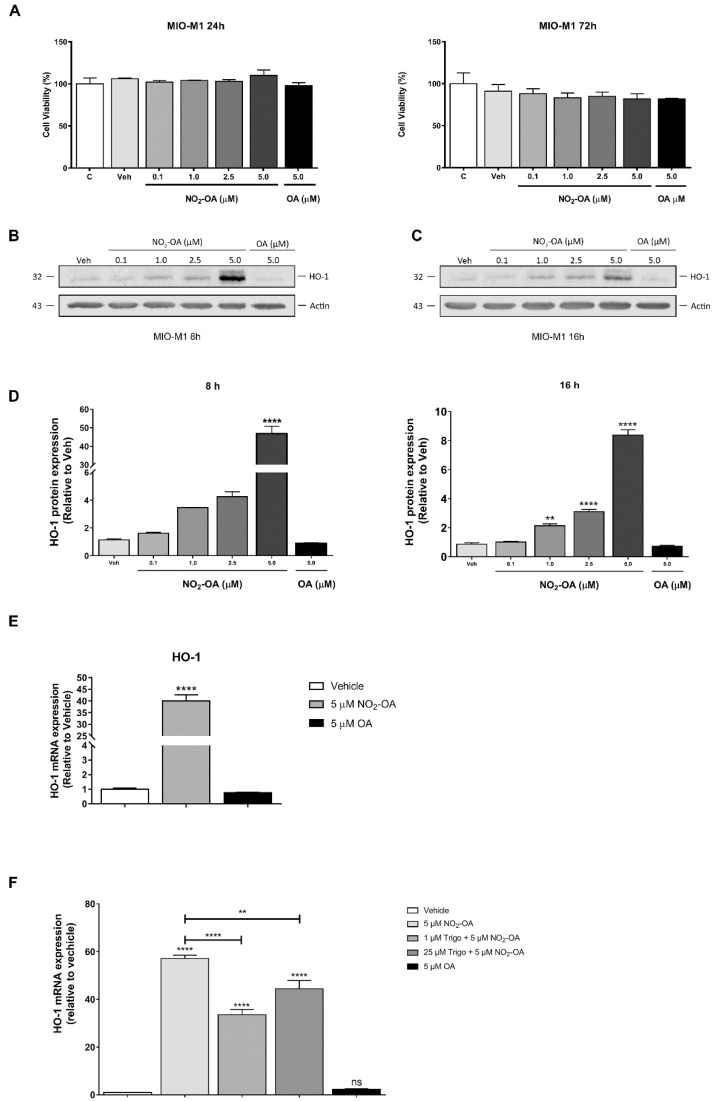
Effect of NO_2_-OA on cell viability and HO-1 expression in MGCs. (**A**) MTT assay represents the percentage of viable cells treated with NO_2_-OA for 24 and 72 h with respect to control. Results are representative of three independent experiments performed in triplicates (means ± SEM). Data were analyzed using a one-way ANOVA followed by Dunnett’s post-test. No statistically significant differences were found in the evaluated conditions. (**B**,**C**) Western blot of HO-1 expression from MIO-M1 cells incubated in the presence or absence of NO_2_-OA (0.1 to 5 µM) for 8 and 16 h. β-actin was used as a loading control. (**D**) Densitometry of HO-1 expression at 8 and 16 h, normalized with β-actin. Results are from three independent experiments. Data are presented as means ± SEM. Data were analyzed using a one-way ANOVA followed by Dunnett’s post-test. ** *p* < 0.01, **** *p* < 0.0001. (**E**) HO-1 mRNA quantification using qRT-PCR in MIO-M1 cells treated with 5 µM NO_2_-OA, vehicle, or OA for 4 h. (**F**) HO-1 mRNA quantification using qRT-PCR in MIO-M1 cells treated with the Nrf2 inhibitor trigonelline. The graph shows the quantification of HO-1 mRNA using qRT-PCR. The results were normalized to GAPDH and expressed according to the 2-ΔΔCt method using the mRNA level obtained from control MIO-M1 cells (vehicle) as a calibrator. Data are presented as means ± SEM and were analyzed using a one-way ANOVA followed by Tukey’s multiple comparison post-test. ns: not significant, ** *p* < 0.01, **** *p* < 0.0001. The results of at least three independent experiments are shown.

**Figure 2 cells-12-00494-f002:**
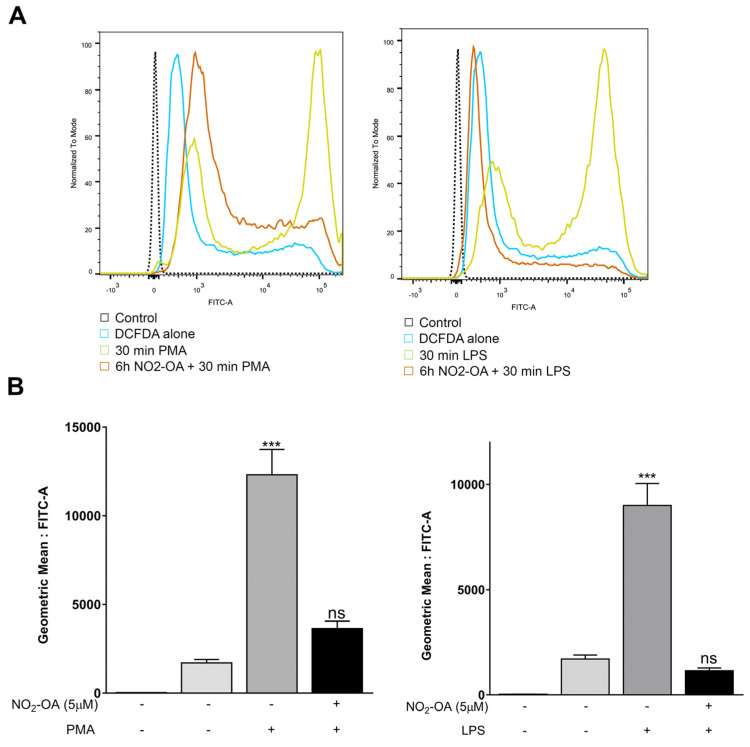
NO_2_-OA prevents the ROS increase induced by LPS or PMA in MGCs. (**A**) NO_2_-OA (5 µM) treatment avoids PMA or LPS induction of ROS increase in MIO-M1 cells. ROS levels were determined with a DCF-DA probe using flow cytometry. (**B**) Data are presented as means ± SEM of the geometric mean and were analyzed using a one-way ANOVA followed by Dunnett’s post-test. ns: not significant; *** *p* < 0.001.

**Figure 3 cells-12-00494-f003:**
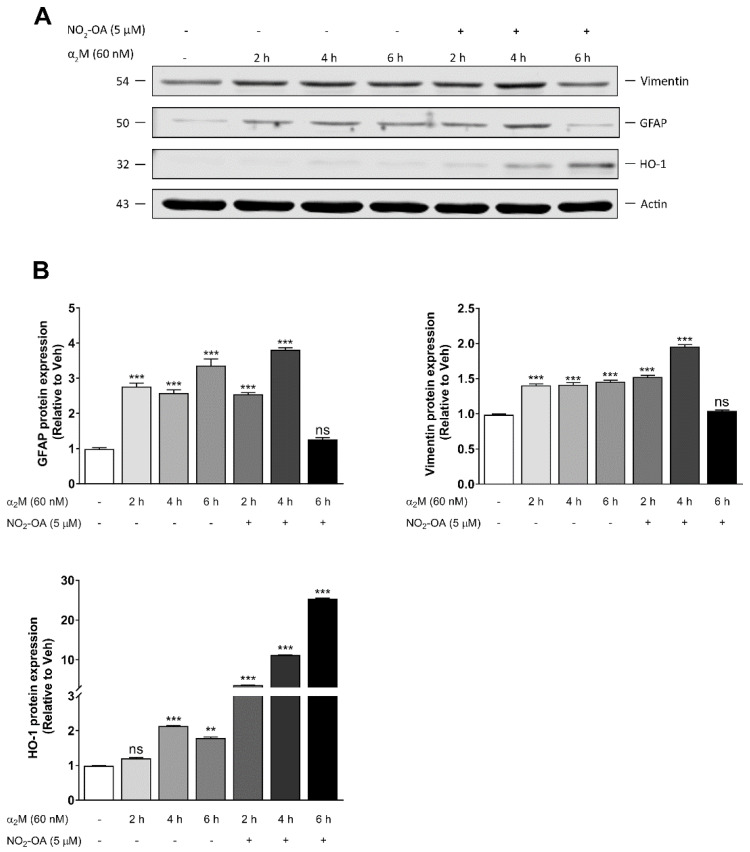
NO_2_-OA reverts glial stress induced by α_2_M* in MGCs. (**A**) Western blot of vimentin, GFAP, and HO-1 in MIO-M1 cells treated in the presence or absence of NO_2_-OA (30 min) before stimulation with α_2_M* for 2, 4, or 6 h. β-actin is shown as a loading control. (**B**) Densitometry of vimentin, GFAP, and HO-1, normalized with β-actin. Data are presented as means ± SEM. Data were analyzed using a one-way ANOVA followed by Dunnett’s post-test. ns: not significant, ** *p* < 0.01, *** *p* < 0.001.

**Figure 4 cells-12-00494-f004:**
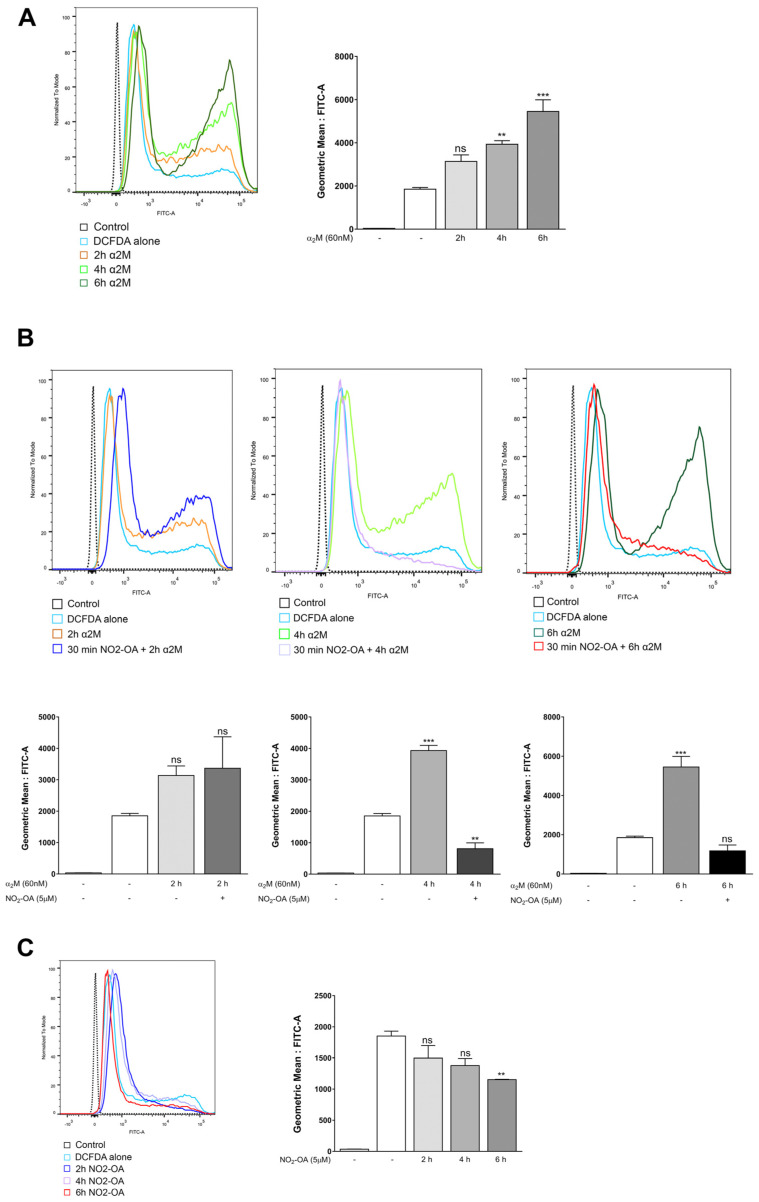
NO_2_-OA inhibits ROS generation induced by α_2_M* in MGCs. (**A**) MIO-M1 cells treated with or without α_2_M* (60 nM) for 2, 4, or 6 h. ROS levels were determined with a DCF-DA probe using flow cytometry. (**B**) MIO-M1 cells were incubated with or without NO_2_-OA (30 min) prior to α_2_M* stimulus for 2, 4, or 6 h. ROS levels were measured with a DCF-DA probe using flow cytometry. (**C**) Evaluation of the NO_2_-OA effect, by itself, on MIO-M1 cells for 2, 4, or 6 h. Data are presented as means ± SEM of geometric means and were analyzed using a one-way ANOVA followed by Dunnett’s post-test. ns: not significant, ** *p* < 0.01, *** *p* < 0.001.

**Figure 5 cells-12-00494-f005:**
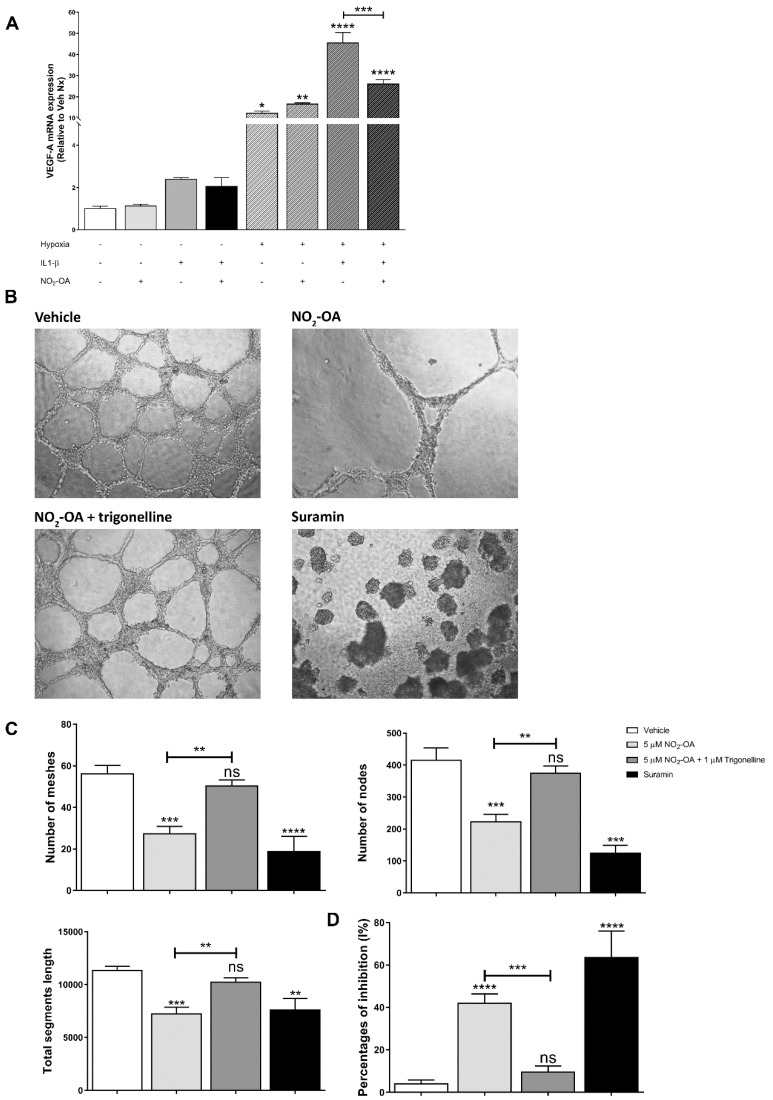
Effect of NO_2_-OA on angiogenesis in MGCs and tube formation in BAECs. (**A**) Expression of VEGF-A mRNA under hypoxic and pro-inflammatory conditions was quantified using qRT-PCR in MIO-M1 cells treated with or without 5 μM of NO_2_-OA and 20 ng/mL IL-1β. The results were normalized to GAPDH and expressed according to the 2-ΔΔCt method using the level of mRNA obtained from the normoxic control as a calibrator. Data are presented as means ± SEM and were analyzed using a one-way ANOVA followed by Tukey’s multiple comparison post-test. * *p* < 0.05, ** *p* < 0.01, *** *p* < 0.001, **** *p* < 0.0001. Results of at least three independent experiments are shown. (**B**) Bright field microscopy images of BAEC (≈1.5 × 10^4^ cells) on Matrigel. Cells were incubated in the presence or absence of 5 µMNO_2_-OA with VEGF (10 ng/mL). Sodium suramin (30 µM) was used as a control of inhibition. Trigonelline (1 µM) was used as an Nrf2 inhibitor. (**C**) Quantitative analysis of the mesh number (polygonal vascular structures), average mesh area, nodes, and total segment length. (**D**) Percentages of inhibition (I%) were calculated as follows: I% = [1 − (Total tube length treatment/Total tube length control)] × 100. Data are presented as means ± SEM. ns: not significant, ** *p* < 0.01, *** *p* < 0.001, **** *p* < 0.0001. The results of at least three independent experiments are shown.

## Data Availability

Not applicable.

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
