# Peer review of "Protective Effect of NO2-OA on Oxidative Stress, Gliosis, and Pro-Angiogenic Response in Müller Glial Cells"

_cells, 2023, doi:10.3390/cells12030494_

Round 1

Reviewer 1 Report

The manuscript by Vaglienti et al investigates the role of nitro-oleic acid (NO2-OA), a nitrated fatty acid, in preventing oxidative stress, gliosis and the pro-angiogenic response in Müller glial cells. These nitrated fatty acids are novel lipid mediators with anti-inflammatory and anti-fibrotic actions. The results are very interesting, since they provide the first evidence that NO2-OA reduces oxidative stress and gliosis in MIO-M1, a Müller glial cell line exposed to different triggers, reduces VEGF increase in a hypoxic/inflammatory context and prevents tubulogenesis in BAEC cells, mimicking an anti-angiogenic effect. They evidence this nitrated fatty acid was not cytotoxic for MIO-M1 cells, which is very relevant for its potential therapeutical use. They also suggest a possible mechanism for NO2-OA action, through the regulation of the Nrf2 pathway and heme oxygenase-1 upregulation, known to provide protection from oxidative stress and inflammation. The experiments are well-designed, with state of the art methodology. The manuscript provides original and relevant data and proposes these fatty acids as potential tools for treating proliferative retinopathies. However, the authors need to provide some additional information to clarify how experiments were performed and they should further discuss their findings to improve the manuscript.

Major points

Abstract: authors, please include in the abstract a brief description of the methodology, indicating the cell lines used (MIO-M1 and BAEC cells), the effect of each of the inductors of damage, to better understand NO2-OA actions.  Please define the abbreviations PMA, LPS and explain trigonelline function, to improve understanding of the data.

Introduction

Pg. 2, 2nd paragraph: “In retina, we have previously demonstrated that α2M* induces gliosis in MGCs at concentrations that were described in the vitreous humor of patients with retinal diseases (9).” Do the authors mean that the α2M* concentrations used were similar to those present in the vitreous humor of patients?

Pg. 2: Have NO2-FA been detected in the eye?

Pg. 2, 7th paragraph: how are MIO-M1 cells affected by PMA and LPS treatment? Why were these different treatments chosen? Please provide some information and the supporting references, to improve clarity and understanding of the purposes of the experiments performed.

Methods

Please describe the NO2-OA treatment: concentrations of NO2-OA used, and how were they solubilized and added to the cultures. Was media replaced after the 30 min incubation with NO2-OA or was the nitrated fatty acid still present in the media during the different treatments?

Please provide information on the assays involving trigonelline and α2M*, which is missing in the Method section. Although α2M* treatment is partially described in the Results section, further information regarding how it was added to the cultures, vehicle used, should be provided. Also indicate the concentrations of PMA and LPS used, how they were added to the cultures and the vehicles used.

1.7: “Sodium suramin 30 μM were used as positive control of inhibition. “ Please indicate sodium suramin function and the type of inhibition (cell proliferation, tube formation) evaluated with it. Why was this concentration chosen? How was the regent solubilized? Please describe in this section how was the VEGF treatment done (concentration used, time of exposure, how it was added to the cultures, vehicle).

Results

Fig. 1: To evaluate whether NO2-OA activates Keap1/Nrf pathway, the authors analyzed the expression of its target gene, HO-1, and show a remarkable increase in HO-1 transcription and expression after NO2-OA treatment. As stated by the authors, the increase in expression was only partially reduced with trigonelline, suggesting “NO2-OA might acts in either Nrf2-dependent and -independent manner on the modulation of HO-1 “. Which might the NRF2-independent pathways or transcription factors be?

Fig. 1E seems to combine the data shown in Fig. 1D. If this is so, just leave one of the figures (1D or 1E), to avoid repeating the information.  Please indicate the numbers in this figure with dots instead of commas (e.g. 0.1, 1.0 instead of 0,1, 1,0). Also replace “actina” by “actin”.

Fig. 3: Were the cells exposed to α2M* and then with NO2-OA, as stated in page 9 or pre-treated with NO2-OA and then with to α2M*, as indicated in the legend of Figure 3? Please clarify.

Please replace “actina” by “actin” in Fig. 3A.

Fig. 5: have the authors checked whether suranim treatment induced cell death? While the reduction in the different parameters is quantitatively similar to that of NO2-OA, cell morphology and formation of structures are markedly different.

Discussion

The Discussion is very brief and could be expanded to provide more context to better understand the relevance and implication of the data.

Page 15, last paragraph: “In addition, a slight increase in HO-1 expression was also observed, which may be associated with the mechanism of induction of gliosis.”  Do the authors mean the mechanism of induction or of reduction of gliosis? Data from Figs. 3 and 4 show that the reduction induced by NO2-OA of vimentin and GFAP expression occurred only after a 6 h exposure to α2M*, while its effect promoting HO-1 increase was visible at shorter times of exposure to α2M*. Could this be related to the reduction in α2M*-induction of gliosis? Have similar results been reported in other models? Authors, please comment further on these findings.

Minor points

There are several typos throughout the text and English grammar should be carefully revised.

Please check the numbering of the subsections, since they start with 1 (e.g., 1.1, 1.2) both in Materials & Methods and Results.

Author Response

Reviewer #1:

We would like to thank Reviewer 1 for their thoughtful comments and useful suggestions. Please find our responses as follows:

Major points

Abstract: authors, please include in the abstract a brief description of the methodology, indicating the cell lines used (MIO-M1 and BAEC cells), the effect of each of the inductors of damage, to better understand NO2-OA actions.  Please define the abbreviations PMA, LPS and explain trigonelline function, to improve understanding of the data.

We thank the reviewer for this comment and we have modified the Abstract in the revised version to improve clarity and understanding.

Introduction

Pg. 2, 2nd paragraph: “In retina, we have previously demonstrated that α2M* induces gliosis in MGCs at concentrations that were described in the vitreous humor of patients with retinal diseases (9).” Do the authors mean that the α2M* concentrations used were similar to those present in the vitreous humor of patients?

The reviewer is right. We have re-written this paragraph for a better understanding as follow:

In retina, we have previously demonstrated that α2M* induces gliosis in MGCs at similar concentrations to those present in the vitreous humor of patients with retinal diseases (9).

Pg. 2: Have NO2-FA been detected in the eye?

Thank you for pointing this out. NO2-FA presence in the eye has not been explored yet. Detection and quantification of free NO2-FA is laborious because they react with thiol-containing proteins present in plasma or cellular membranes. In humans, NO2-FAs have been described and quantified in plasma and urine by mass spectrometry. We are interested in evaluating local concentration of NO2-FA, however, in this study our aim was to explore the action of NO2-OA in the retina under pathological conditions. 

Pg. 2, 7th paragraph: how are MIO-M1 cells affected by PMA and LPS treatment? Why were these different treatments chosen? Please provide some information and the supporting references, to improve clarity and understanding of the purposes of the experiments performed.

In order to clarify this point: in the retina, there are two cell types that belong to macroglia, one of them is MGCs. These cells expressed LPS receptor, Toll like 4 (TLR4) and play an important role in local immune response supporting neuron health and activity. LPS and PMA represent a classical danger signal for the immune system which activates the defense system in immune innate cells. For this reason, in this study, we have explored the action of NO2-FA in MGCs activated with LPS and PMA. As a proof of concept, it was recently described that LPS induces the participation of MGC in the neuroinflammatory process in the retina (Lorenz, et al., Cells 2021 Mar 23;10(3):711. doi: 10.3390/cells10030711).

Methods

Please describe the NO2-OA treatment: concentrations of NO2-OA used, and how were they solubilized and added to the cultures. Was media replaced after the 30 min incubation with NO2-OA or was the nitrated fatty acid still present in the media during the different treatments?

We have added the following description for NO2-OA treatment in the Methods section:

NO2-OA were solubilized in methanol and added in a 1:1000 dilution in cell culture media (DMEM) for all treatments. MIO-M1 cells were treated with vehicle (methanol) or different concentrations of NO2-OA (0.1, 1.0, 2.5, 5.0 µM) for 8 or 16 h. On the other hand, MIO-M1 cells were incubated in presence or absence of 5 µM of NO2-OA for 30 min before adding α2M* (60 nM) for 2, 4 and 6 h.

Please provide information on the assays involving trigonelline and α2M*, which is missing in the Method section. Although α2M* treatment is partially described in the Results section, further information regarding how it was added to the cultures, vehicle used, should be provided. Also indicate the concentrations of PMA and LPS used, how they were added to the cultures and the vehicles used.

We have added information about the assays involving LPS, PMA and trigonelline in the following Methods subsection. α2M* treatment information was answered above.

2.5. Dichlorofluorescein assay

Briefly, MIO-M1 cells were treated with NO2-OA (5 μM) or vehicle (methanol) for 6 h before adding PMA (1µg/ml) or LPS (1 nM) for 30 min.)

2.8 Tube Formation Assay

The plates were incubated for 18 h (37 °C, 5% CO2) in the presence and absence of NO2-OA (5 µM) and trigonelline (1 µM) with the addition of VEGF (10 ng/mL) as angiogenic stimuli. NO2-OA and trigonelline stock were prepared in DMSO and PBS, respectively. Recombinant VEGF was reconstituted in distilled water to a concentration of 0.1 mg/ml. Suramin sodium 30 µM was used as positive inhibition control. The stock solution of Suramin sodium was prepared in ethanol. Controls with or without vehicles (DMSO or ethanol) were simultaneously run. The final concentration of vehicle per well in this and all the experiments described herein was 0.1% v/v. No adverse effects were observed at this concentration.

1.7: “Sodium suramin 30 μM were used as positive control of inhibition. “Please indicate sodium suramin function and the type of inhibition (cell proliferation, tube formation) evaluated with it. Why was this concentration chosen? How was the regent solubilized? Please describe in this section how was the VEGF treatment done (concentration used, time of exposure, how it was added to the cultures, vehicle).

We agree with the reviewer’s comments. We have included the information required to suramin and VEGF in the answer above.

The exact mechanism of action of suramin is still unknown, but suramin has been shown to inhibit angiogenesis by blocking specific receptor binding of PDGF, bFGF, VEGF and neuropilin and might have been expected to show a cluster morphology (Weisz et al., Circulation 2001, 103, 1887–1892; Gagliardi et al.,  Cancer Res. 1992, 52, 5073–5075; Cook, et al., Lung Cancer 2006, 42, 263–274).

In addition, Arlinda et al. described that 10 µM and 50 µM of suramin showed an inhibitory effect on cellular network formation, by forming cords with interactions, and a complete absence of network at 100 µM, by forming only cords (Ljoki A et al.,  Int J Mol Sci. 2022 Apr 12;23(8):4277. doi: 10.3390/ijms23084277. PMID: 35457095; PMCID: PMC9025250). According to these data we have selected for this and other papers (Subirada et al., Front. Cell Dev. Biol., Febrero 14, 2022 doi: 10.3389/fcell.2022.855178) the dose of 30 μM.

Results

Fig. 1: To evaluate whether NO2-OA activates Keap1/Nrf pathway, the authors analyzed the expression of its target gene, HO-1, and show a remarkable increase in HO-1 transcription and expression after NO2-OA treatment. As stated by the authors, the increase in expression was only partially reduced with trigonelline, suggesting “NO2-OA might acts in either Nrf2-dependent and -independent manner on the modulation of HO-1 “. Which might the NRF2-independent pathways or transcription factors be?

We agree with the reviewer that induction of HO-1 should be explained in more detail. To do that we rewrite the mentioned paragraph in the Discussion section: “NO2-OA might act in either Nrf2-dependent and -independent manner on the modulation of HO-1. This result may be attributed to previous work from Wright et al, which showed transcriptional modulation of HO-1 by NO2-LA (nitro linoleic acid) through cAMP, AP-1 and E-box response element interaction in MEF and human aortic endothelial cells (Wright et al., Biochem J. 2009 Aug 13;422(2):353-61. doi: 10.1042/BJ20090339.)”.

Fig. 1E seems to combine the data shown in Fig. 1D. If this is so, just leave one of the figures (1D or 1E), to avoid repeating the information.  Please indicate the numbers in this figure with dots instead of commas (e.g. 0.1, 1.0 instead of 0,1, 1,0). Also replace “actina” by “actin”.

We thank the reviewer for this observation. So, we have eliminated Fig 1E to avoid duplication of the information and we have corrected the numbering and misspelling (actin).

Fig. 3: Were the cells exposed to α2M* and then with NO2-OA, as stated in page 9 or pre-treated with NO2-OA and then with to α2M*, as indicated in the legend of Figure 3? Please clarify.

Please replace “actina” by “actin” in Fig. 3A.

We thank reviewer for this comment, and we have now re-written this paragraph improving its understanding as follow:

In order to evaluate whether NO2-OA abrogate glial stress in MGCs, we incubated MIO-M1 cells in presence or absence of 5 µM of NO2-OA for 30 min before α2M* (60 nM) treatment for 2, 4 and 6 h.

In addition, we have replaced “actina” by actin in Fig. 3A.

Fig. 5: have the authors checked whether suranim treatment induced cell death? While the reduction in the different parameters is quantitatively similar to that of NO2-OA, cell morphology and formation of structures are markedly different.

We thank the reviewer for this comment. By MTT assays we observed that suramin sodium did not significantly affect BAEC cell viability at this concentration. We checked this effect and we have worked with this dose in different papers already published by our lab (Subirada et al., Front. Cell Dev. Biol., 2022; Llorens de los Ríos et al., Front. Pharmacol. 2022, 13:1007790. doi: 10.3389/fphar.2022.1007790).

As was mentioned above, NO2-OA and suramin exhibit antiangiogenic properties mediated by different mechanisms, which impact in a different way on cell morphology and formation of structures. 

Discussion

The Discussion is very brief and could be expanded to provide more context to better understand the relevance and implication of the data.

We thank the reviewer for this observation, and we have now modified the Discussion section adding comments to specific points required by the reviewer. We consider to be prudent and careful in the discussion of our results.

Page 15, last paragraph: “In addition, a slight increase in HO-1 expression was also observed, which may be associated with the mechanism of induction of gliosis.”  Do the authors mean the mechanism of induction or of reduction of gliosis? Data from Figs. 3 and 4 show that the reduction induced by NO2-OA of vimentin and GFAP expression occurred only after a 6 h exposure to α2M*, while its effect promoting HO-1 increase was visible at shorter times of exposure to α2M*. Could this be related to the reduction in α2M*-induction of gliosis? Have similar results been reported in other models? Authors, please comment further on these findings.

To clarify the point brought up by the reviewer and be more specific, we have reworded the paragraph in the manuscript:

“In addition, a slight increase in HO-1 expression was also observed, which may be associated with induction of ROS and gliosis by α2M.”

Regard to the following question point it by the reviewer: Could this be related to the reduction in α2M*-induction of gliosis?

We believe that induction of HO-1, via Nrf2-dependent or -independent, is involved in the reduction of gliosis and ROS by NO2-OA. HO-1 is an inflammatory mediator that participates in the catabolism of hemoglobin to biliverdin, carbon monoxide (CO) and ferric iron and the sub-products of the haem have been described to display important anti-inflammatory properties. In addition, we have explored the modulation of Keap1/Nrf2 pathway by NO2-FA in numerous cellular and animal models, and point out that the activation of this pathway plays a leading role in the cytoprotective and antioxidant properties of NO2-FA (Kansanen et al., J Biol Chem. 2011 Apr 22;286(16):14019-27. doi: 10.1074/jbc.M110.190710. Epub 2011 Feb 25; Khoo et al., Sci Rep. 2018 Feb 2;8(1):2295. doi: 10.1038/s41598-018-20460-8).

Thus, the dynamic of HO-1 expression (Fig 3: start at 2 h to reach maximum at 6 h) contributes to improved gliosis markers after 6 h (GFAP and vimentin) by effect of NO2-FA.    

Minor points

There are several typos throughout the text and English grammar should be carefully revised.

The revised version has been carefully reviewed by a native English-speaking colleague.

Please check the numbering of the subsections, since they start with 1 (e.g., 1.1, 1.2) both in Materials & Methods and Results.

We are thankful for this observation and we have corrected them in the revised version.

Reviewer 2 Report

Minor:

1. The author need to highlight why they selected only VEGF as angiogenic factors and not other related factors involved?

2. The future directions regarding exploring the effect of  NO2-OA on other components other than VEGF need to be incorporated. 

3. Study limitations need to be mentioned.

Author Response

Reviewer #2:

  1.     The author need to highlight why they selected only VEGF as angiogenic factors and not other related factors involved?

We thank the reviewer for this observation. Although proliferative neovascular retinopathies are caused by several growth factors (Wells et al., N Engl J Med. 2015; 372: 1193-203; Durham and Herman, Curr Diab Rep. 2011; 11: 253-64), it is widely known that VEGF-A is the most potent cytokine that mediates ischemia-induced retinal neovascularization (NV) in ocular pathologies (Aiello LP. Ophthalmic Res. 1997; 29: 354-62; Das A and McGuire PG. Prog Retin Eye Res. 2003; 22: 721-48). VEGF-A exerts its effects via activation of various signaling events, including tyrosine phosphorylation of its receptors VEGFR1 (Flt1), VEGFR2 (KDR/Flk-1), and VEGFR3 (Flt-4) and their downstream effectors on endothelial cells (ECs) (Chung AS and Ferrara N. Annu Rev Cell Dev Biol. 2011; 27: 563-84). Accumulating evidences, stemming from experimental models and clinical studies have provided insights into the mechanisms of vascular injury leading to pathological vitreoretinal NV and implementation of ocular anti-VEGF therapies (Tah et al., J Ophthalmol. 2015; 2015: 627674). Bevacizumab (Avastin®; Genentech, San Francisco, CA), ranibizumab (Lucentis®; Genentech, San Francisco, CA), and aflibercept (Eylea®; Regeneron Pharmaceuticals, Tarrytown, NY) are the most widely used anti-VEGF drugs administered by intravitreal injection in ocular pathologies.  

Given the reviewer's comment, we have added a paragraph in the Discussion section (pag 16):

Accumulating evidences, stemming from experimental models and clinical studies have provided insights into the mechanisms of vascular injury leading to pathological vitreoretinal NV and implementation of ocular anti-VEGF therapies (Tah et al., J Ophthalmol. 2015; 2015: 627674). Bevacizumab (Avastin®; Genentech, San Francisco, CA), ranibizumab (Lucentis®; Genentech, San Francisco, CA), and aflibercept (Eylea®; Regeneron Pharmaceuticals, Tarrytown, NY) are the most widely used anti-VEGF drugs administered by intravitreal injection in ocular pathologies.

  1.     The future directions regarding exploring the effect of  NO2-OA on other components other than VEGF need to be incorporated. 

For several years, our lab has been studying the participation of different proteins and growth factors in the physiological development and pathology of the retina (Sánchez et al., Exp. Eye Res. 2006, 2007, Barcelona et al., Exp. Eye Res. 2010, FASEB Journal 2013, Invest Ophthalmol Vis Sci. 2011) specifically in non-proliferative (Paz et al., Front. Cell Dev. Biol., 2020) and proliferative retinopathies. Among them, we have reported the contribution of IGF-1/IGF-1R system (Lorenc et al., Invest Ophthalmol Vis Sci. 2015, Mol Neurobiol. 2017) and Gal1 (Ridano et al., Oncotarget. 2017) to vascular and non-vascular alterations. However, in this work we just explore the NO2-OA capacity to regulate VEGF-A in MGCs. We agree with the reviewer that this topic would be interesting to evaluate in future research with focus on other factors such as IGF-1 and Gal-1, among others.

  1.     Study limitations need to be mentioned.

In agreement with the reviewer, our experimental model has limitations to explain pathophysiological events that occur in the retina during retinopathies. Therefore, in the Discussion Section we have included the following paragraph: Additional experiments are in progress in an oxygen induced-retinopathy mouse model to highlight clinical and therapeutic connotations of our in vitro results in neovascular and neurodegenerative disorders of the retina. 

Reviewer 3 Report

could you please explain more how did you measure ROS level?

Author Response

Reviewer #3:

Could you please explain more how did you measure ROS level?

Recently, we have described a detailed and reproducible protocol to measure and quantify ROS levels by flow cytometry in MGCs (Vaglienti et al., J Vis Exp. 2022 May 13;(183). doi: 10.3791/63337). In this protocol, we provide a complete description of the procedures required for the measurement of ROS with DCFH-DA probe and flow cytometry in MGCs. Quantifying ROS levels with a reproducible and simple method is essential to assess the contribution of pathways or molecules that participate in antioxidant cell defense mechanisms.